# Hypoxic and Hyperoxic Breathing as a Complement to Low-Intensity Physical Exercise Programs: A Proof-of-Principle Study

**DOI:** 10.3390/ijms22179600

**Published:** 2021-09-04

**Authors:** Costantino Balestra, Kate Lambrechts, Simona Mrakic-Sposta, Alessandra Vezzoli, Morgan Levenez, Peter Germonpré, Fabio Virgili, Gerardo Bosco, Pierre Lafère

**Affiliations:** 1Environmental, Occupational, Aging (Integrative) Physiology Laboratory, Haute Ecole Bruxelles-Brabant (HE2B), 1180 Brussels, Belgium; klambrechts@he2b.be (K.L.); mlevenez@he2b.be (M.L.); pgermonpre@gmail.com (P.G.); plafere@he2b.be (P.L.); 2Physical Activity Teaching Unit, Motor Sciences Faculty, Université Libre de Bruxelles (ULB), 1050 Brussels, Belgium; 3DAN Europe Research Division, 1160 Brussels, Belgium; 4Institute of Clinical Physiology, National Research Council (IFC-CNR), Piazza dell’Ospedale Maggiore, 3, 20162 Milan, Italy; simona.mrakicsposta@cnr.it (S.M.-S.); alessandra.vezzoli@cnr.it (A.V.); 5Centre for Hyperbaric Oxygen Therapy, Queen Astrid Military Hospital, 1120 Brussels, Belgium; 6Council for Agricultural Research and Economics—Food and Nutrition Research Centre (C.R.E.A.-AN), 00178 Rome, Italy; 7Environmental Physiology & Medicine Lab, Department of Biomedical Sciences, University of Padova, 35131 Padova, Italy

**Keywords:** normobaric oxygen paradox, hypoxia, hyperoxia, physical activity, inflammation, immune response, oxidative stress, rehabilitation

## Abstract

Inflammation is an adaptive response to both external and internal stimuli including infection, trauma, surgery, ischemia-reperfusion, or malignancy. A number of studies indicate that physical activity is an effective means of reducing acute systemic and low-level inflammation occurring in different pathological conditions and in the recovery phase after disease. As a proof-of-principle, we hypothesized that low-intensity workout performed under modified oxygen supply would elicit a “metabolic exercise” inducing a hormetic response, increasing the metabolic load and oxidative stress with the same overall effect expected after a higher intensity or charge exercise. Herein, we report the effect of a 5-week low-intensity, non-training, exercise program in a group of young healthy subjects in combination with the exposure to hyperoxia (30% and 100% pO_2_, respectively) or light hypoxia (15% pO_2_) during workout sessions on several inflammation and oxidative stress parameters, namely hemoglobin (Hb), redox state, nitric oxide metabolite (NOx), inducible nitric oxide synthase (iNOS), inflammatory cytokine expression (TNF-α, interleukin (IL)-6, IL-10), and renal functional biomarkers (creatinine, neopterin, and urates). We confirmed our previous reports demonstrating that intermittent hyperoxia induces the normobaric oxygen paradox (NOP), a response overlapping the exposure to hypoxia. Our data also suggest that the administration of modified air composition is an expedient complement to a light physical exercise program to achieve a significant modulation of inflammatory and immune parameters, including cytokines expression, iNOS activity, and oxidative stress parameters. This strategy can be of pivotal interest in all those conditions characterized by the inability to achieve a sufficient workload intensity, such as severe cardiovascular alterations and articular injuries failing to effectively gain a significant improvement of physical capacity.

## 1. Introduction

Inflammation is an important adaptive response to both external and internal stimuli including infection, trauma, surgery, ischemia-reperfusion, and malignancy. It is a normal body function, primarily aiming at an adaptational response to abnormal conditions and a restoring of homeostasis [1,2]. One of the most evident features of acute inflammation is the expression and release of proinflammatory cytokines such as tumor necrosis factor-alpha (TNF-α), interleukin-1β (IL-1β), interleukin-6 (IL-6), and the expression of cyclooxygenase (COX), an enzyme implicated in the synthesis of prostaglandins (PGs) and therefore in the transition to and the maintenance of chronic inflammation [3]. This cascade of events, when unregulated, is widely considered an important pathogenic factor contributing to chronic-degenerative diseases including cardiovascular diseases, diabetes, Alzheimer’s, rheumatoid arthritis, chronic obstructive pulmonary diseases, some specific types of cancer, chronic kidney diseases or obesity [4,5]. In the presence of unregulated inflammatory response, proinflammatory and anti-inflammatory pathways are leading to a complex pattern of systemic activations consequent to cytokine storm, resulting in immune system impairment and proinflammatory imbalance. Downstream, the resulting “massive” inflammatory cascade, is responsible for global autonomic, endocrine, hematological, and immunological alterations, leading to a reversible or irreversible end-organ dysfunction and even death, as we have dramatically seen in the course of the current pandemic of severe acute respiratory syndrome corona virus 2 (SARS CoV-2) [6,7].

Thus, an appropriate control of inflammatory response is a requisite for organism’s health. Therefore, when endogenous homeostatic regulation systems fail, anti-inflammatory drugs play a fundamental role. Many, if not all, available anti-inflammatory drugs bring along deleterious side effects such as gastro-intestinal tissue damage or even cardiovascular and kidney toxicity [8]. This prompts the development of alternative interventions to prevent and/or reduce uncontrolled inflammatory response. Although low-intensity, short-term physical activity does not cause significant changes in immunological and physiological responses [9], studies dealing with the role of exercise [10,11,12,13] and relative hypoxic stress exposure [14] demonstrated that muscle-damaging exercise triggers a delayed systemic, predominantly anti-inflammatory, acute cytokine response.

The importance of skeletal muscle mass and strength therapeutic targets in medicine is hindered in patients who do not have the possibility to achieve a sufficient training intensity to effectively gain a significant improvement of physical capacity. Moreover, effective resistance training to increase muscle mass and strength could worsen disease conditions in patients with severe cardiovascular alterations and injuries [15]. A technique relying on a circulatory occlusion in resistance training combined with a very low-intensity mechanical load has been proven effective to enhance muscular metabolic stress, thereby increasing muscle mass. This technique is based on the idea that cellular hypoxia can trigger the expression of several inflammatory mediators which signal tissue damage and initiate a survival hormetic response. This response to hypoxia is controlled by the hypoxia-inducible factor-alpha (HIF-1α) signaling cascade [16]. 

Even though mainly activated by hypoxia, TNF-α and interleukin (IL)-18 have also been reported to be associated with an increased expression of HIF-dependent genes via crosstalk with the NF-κB pathway [15,17,18].

Finally, we have demonstrated [17,19,20] that intermittent hyperoxia induces HIF-1α activation and the expression of genes involved in the response to low oxygen. This response, known as “normobaric oxygen paradox” (NOP), describes the effects to the return to normoxia after a relatively short hyperoxic event, which is sensed by tissues as oxygen shortage. This phenomenon nowadays has several different names either “hyperoxic-hypoxic paradox” [21] or “normobaric oxygen paradox” [22], depending on the range of variation of pO_2_ imposed. Nevertheless, a general term could be a “relative hypoxia” without reaching hypoxic levels within organism’s tissues [22].

In order to demonstrate the feasibility of the application of NOP [23] to optimize a workout program specifically addressed to impaired patients, we designed a proof of principle study. Accordingly, we hypothesized that low-intensity workout carried out under increased partial pressure of oxygen (pO_2_) might trigger a response overlapping an exercise performed at higher intensity or charge. We assumed, in fact, that a program based on low-intensity workout under modified oxygen supply would elicit a sort of “metabolic exercise” inducing a hormetic response leading to an increase of the metabolic load and oxidative stress with the same overall effect expected after an exercise performed at higher intensity or charge in all those conditions where heavy muscular work is not achievable.

Herein, we reported the results of a proof-of-concept study investigating the effect of a 5-week, low-intensity exercise program in a group of healthy young subjects in combination with the exposure to mild or high hyperoxia or light hypoxia during workout sessions on several inflammation and oxidative stress parameters, namely hemoglobin (Hb), nitric oxide metabolite (NOx), inducible nitric oxide synthase (iNOS), inflammatory cytokine expression (TNF-α, IL-6, and IL-10), and renal functional biomarkers (creatinine, neopterin, and urates).

## 2. Results

### 2.1. Hypoxia and High Hyperoxia Induce the Hb Synthesis Similar to an Hypoxic Stimulus

As mentioned in the introduction, we have previously demonstrated that the administration of intermittent hyperoxia induces a “paradoxical” effect, largely overlapping the physiological response to hypoxia, characterized by HIF-1α activation in cultured cells and in vivo, in humans [19,20,24]. In the present study, we confirmed, within a different experimental frame, that repeated short exposures to a light hypoxia and high hyperoxia induce a significant increase of the Hb synthesis, reaching values of 117.3% ± 17.6% (*p* = 0.0005; one sample *t*-test) and 120.8% ± 23.1% (*p* = 0.01; one sample *t*-test) in subjects breathing 15% and 100% O_2_, respectively. Conversely, the subjects exercising while breathing a mildly hyperoxic air mixture (30% O_2_) did not show any significant increase of the Hb synthesis within the experimental study window (Figure 1, panel A). The longitudinal time evolution of the hematocrit throughout the study followed and paralleled the increase of Hb levels (Figure 1, panel B). As expected, the exercise routine performed at normoxic pO_2_ had no effect on both Hb levels and hematocrit.

### 2.2. Low-Level Exercise Performed under Hypoxia or Hyperoxia Is Associated with an Increased Production of Reactive Oxygen Species (ROS) along with a Decreased Total Antioxidant Capacity (TAC)

An enhanced generation of ROS and a parallel decrease of the plasmatic antioxidant parameters such as the total antioxidant capacity (TAC) is one of the most frequently reported features of the exposure to a hypoxic or hyperoxic environment. Alterations of the redox balance induces, in turn, an adaptation mediated by the activation of hormetic response pathways such as those depending on nuclear factor (erythroid-derived 2)-like 2 (NRF2) and NF-κB transcription factors [25]. At the end of the 5-week exercise program, we observed (Figure 2A) significant increases, with respect to the baseline value, of the plasmatic ROS production rate equal to 134.9% ± 25.6% (*p* = 0.031), 149.7% ± 62.1% (*p* = 0.004), and 145.2% ± 26.8% (*p* = 0.031), for 15%, 30%, and 100% O_2_ mixtures, respectively. This increase was accompanied by a decrease of plasmatic TAC, with respect to the baseline value, in subjects performing their workout sessions at 30% and 100% O_2_ (89.4% ± 8.9% (*p* = 0.001) and 86.3% ± 10.9% (*p* = 0.031), respectively). On the other hand, TAC values remained unchanged in subjects performing while breathing slightly hypoxic air (Figure 2B). The implementation of the exercise program under normal pO_2_ had no significant effect on both markers.

### 2.3. Physical Exercise Performed under Modified Oxygen Concentration Affects iNOS Enzyme Expression and NOx Metabolites Levels

The inducible form of nitric oxide synthase (iNOS) plays an important role in inflammation. Its expression [26] is induced by the activation of transcription factors such as interferon regulatory factor 1 (IRF1)- and NF-κB-associated with inflammation oxidative stress [27]. iNOS produces large quantities of NO upon stimulation, such as by proinflammatory cytokines (e.g., IL-1, TNF-α, and interferon-γ) as an immune defense mechanism and as a vasoconstriction response on peripheral and pulmonary vascular tone modulation.

Figure 3 (panel A) shows that the exercise under modified oxygen concentrations, either hypoxic or mild-high hyperoxic, was paralleled by a downregulation of iNOS enzyme expression (93.3% ± 5.8% (*p* = 0.031), 88.7% ± 6.3% (*p* = 0.004), and 83.6% ± 11.5% (*p* = 0.031) of the baseline value, respectively) apparently associated with the intensity of the alteration. As shown in Figure 3 (panel B), these changes resulted in lower nitrite and nitrate (NOx) plasma levels equal to 65.2% ± 26.5% (*p* = 0.031), 68.6% ± 24.7% (*p* = 0.004), and 64.5% ± 28.2% (*p* = 0.031) of the baseline value, for 15%, 30%, and 100% O_2_ in breathing gas, respectively. The completion of the exercise program in normoxic conditions did not affect these parameters.

### 2.4. Low-Intensity Physical Excercise in Modified Oxygen Concentration Modulates the Inflammatory State

Different forms of physical activity have been reported to stimulate cytokine release from various tissues to the bloodstream. In fact, receptors for exercise-induced cytokines are present in several body tissues including muscle tissue, adipose tissue, and brain [28]. We observed a slight but significant modulation of the level of inflammation-related cytokines in association with all altered oxygen concentration inspired during exercise sessions.

In our experimental frame, IL-6 levels were found significantly affected only by the exposure to mild hyperoxia with 30% of O_2_, during physical exercise (296.2% ± 187.8% with respect of the baseline value; *p* = 0.004; Figure 4, panel A).

At the end of the 5-weeks exercise protocol, the levels of the anti-inflammatory cytokine-10 (IL-10) increased up to 365.8% ± 246.4% (*p* = 0.031), 162.7% ± 70.3% (*p* = 0.014), and 296.2% ± 187.8% (*p* = 0.031) in comparison to the baseline value, in subjects breathing 15%, 30%, and 100% O_2_, respectively. (Figure 4, panel B).

Consistently with that of IL-6, the releases of the proinflammatory cytokine TNF-αwere significantly increased up to 134.4% ± 28.9% (*p* = 0.001) and to 158.9% ± 28.9% (*p* = 0.031) in comparison to the baseline value in subjects breathing 30% and 100% O_2_ during exercise, respectively. Conversely, TNF-α was not significantly affected by exercise performed in mild hypoxia (Figure 5, panel C). Similarly, when the workout program was performed at normal pO_2_, no significant changes of cytokines expression were observed in comparison with levels assessed at the beginning of the exercise program.

### 2.5. A 5-Week Mild Exercise Plan under Altered Oxygen Concentrations Affects Functional Kidney Biomarkers

Creatinine, neopterin, and urates are widely accepted as robust biomarkers of renal function [29]. They have been reported to be sensitive to physical exercise, especially aerobic, in association to a mitigation of adverse effects of sedentary behavior, hypertension, and other pathologies [30].

Figure 5 (panel A) shows that creatinine levels were significantly increased by breathing 15%, 30%, and 100% O_2_ (200.5% ± 77.2% (*p* = 0.031), 174.4% ± 62.1% (*p* = 0.006), and 416.7% ± 227.1% (*p* = 0.017) of the baseline value, respectively).

Differently, subjects breathing 30% and 100% O_2_ during their exercise sessions (panels B and C) presented significantly altered plasmatic levels of neopterin (131.6% ± 26.7% (*p* = 0.002) and 186.0% ± 77.8% (*p* = 0.002)) and urate (177.8% ± 42.4% (*p* = 0.031) and 310.1% ± 105.9% (*p* = 0.031)) in comparison to the background levels. Conversely, breathing a slightly hypoxic air during exercise was not associated with significant changes of plasmatic levels of both neopterin and urate. Similarly, exercising at normal pO_2_ was not associated with significant variations of renal functional parameters in comparison to the initial values.

### 2.6. Mild Exercise, Independently of Administered pO_2_, Does Not Affect the Maximal Oxygen Uptake although Hypoxia Significantly Affects Borg’s Ratings of Perceived Exertion

The 5-week schedule of low-intensity workout, performed at alternate days, had no significant effect on the maximum aerobic capacity overall (41.8 ± 11.9 mL·(kg·min)^−1^ vs. 41.7 ± 12.2 mL·(kg·min)^−1^ at the beginning and at the end of the program, respectively), confirming that the exercise program did not induce a real “training”. Accordingly, the respiration exchange rate (RER) and the maximal workload at the anaerobic threshold were not significantly affected by altered oxygen concentration during exercise (data not shown).

Figure 6 shows the longitudinal variation of the perceived exertion rated after each exercise session according to the Borg CR10 scale. The Borg rating of the perceived exertion scale (RPE scale) is a frequently used quantitative measure of perceived exertion during physical activity, useful to assess the subjective intensity of a workout session. In medicine, this is considered a reliable marker for clinical diagnosis of breathlessness and dyspnea, chest pain, angina, and musculo-skeletal pain [31].

In comparison to initial values, no significant differences in Borg’s score were recorded in subjects performing their exercise under mild (30% O_2_) or high (100% O_2_) hyperoxia (1.45 ± 0.7 vs. 1.9 ± 0.8, respectively (*p* = 0.12)). Conversely, exercise performed under hypoxia (15%) was associated to an apparent modification of the perceived exertion score, being significant from the sixth to the 13th workout sessions (2.9 ± 1.1; *p* < 0.05). Breathing normoxic air during exercise sessions did not affect Borg’s score.

## 3. Discussion and Conclusions

The study presented herein aimed to confirm that a low-intensity, 5-week-long workout program could be an expedient approach effective to modulate the expression of several specific inflammatory mediators and the overall inflammatory status in a group of healthy subjects. Moreover, we considered the administration of exercise protocols together with the exposure to modified air composition, namely slightly hypoxic (15% O_2_) and mildly or frankly hyperoxic (30% and 100% O_2_, respectively). In fact, we started from previous reports indicating that light hypoxia triggers the expression of several inflammatory mediators and initiates a survival hormetic response. Similarly, our group and others demonstrated that the return to normoxia after an intermittent administration of hyperoxic events results in the upregulation of the HIF-1α transcription factor activity and, at the systemic level, in a set-up of the organism adapted to an anti-inflammatory profile [32,33]. More recently, we reported the activation of oxygen-sensitive transcription factors in human peripheral blood mononuclear cells (PBMCs) in healthy subjects after the exposure to a mild or high hyperoxia (30% and 100% O_2_, respectively). Our study indicated that the exposure to mild hyperoxia, perceived as a hypoxic stress, is characterized by the activation of HIF-1α and NRF2, but not nuclear factor kappa-light-chain-enhancer of activated B cells (NF-κB) in circulating PBMCs. In the same study, we also observed that, conversely, a temporary exposure to high hyperoxia leads to a progressive loss of NOP response in favor of features characterizing the presence of a frank oxidative stress associated to the activation of NRF2 and NF-κB and to glutathione synthesis, describing a hormetic adaption of antioxidant cellular defense [17]. These specific transcription factors are not only involved in the hormetic organism’s response to oxidative stress, but also play a pivotal role in inflammation as they control the expression of specific mediators [27]. Therefore, their patterns of activation associated to the extent of oxygen concentration, either hypoxic or mildly/highly hyperoxic, justify the complex picture, resulting from the results presented herein.

In the present study, we confirmed that intermittent exposure to mild or high hyperoxia induces a NOP effect, associated with increased Hb and hematocrit. This is consistent with erythropoietin increase after hypoxia and the already demonstrated normobaric oxygen paradox that leads to increased Hb after repeated exposures as a reaction to HIF-1α activation [22,23,34,35]. A first original result to be considered is in that our study design, based on a 5-week mild workout session on alternate days, was not associated to any significant training effect as indicated by the maintenance of an unaltered RER, independently of the breathing gas composition provided to subjects while performing their exercise. Beyond the confirmation of the NOP effect of high hyperoxic treatment resulting in a significant increase of Hb levels (Figure 1) similar to that of hypoxia, the most evident observed effect is the possibility to induce a significant modulation of inflammatory parameters. This is consistent with recent observation related to the COVID-19 pandemic. Indeed, HIF-1α which activates ARDS-induced hypoxia plays a crucial role in the pathogenesis of cytokine storm, since the expression of TNF-α, IL-1β, and IL-6 which are key elements of cytokine storm by NF-κB [36]. As expected, breathing both mild and high hyperoxic air induced a significant increase of the generation of ROS. Interestingly, also the exposure to hypoxic pO_2_ was associated with a significant increase of ROS generation (Figure 2, panel A). Even though this observation was not expected, there are several reports indicating that hypoxia induces a dysregulation of cellular oxygen metabolism and increased generation of ROS and reactive nitrogen species [37]. Accordingly, TAC, which mainly relates to the availability of small-molecular-weight antioxidants such as protein and non-protein thiols, nutritional, nucleophiles (polyphenols, phenolic acids, isothiocyanates, etc.) rather than enzymatic reduction [38,39] is negatively affected by both mild and high hyperoxia, but not by light hypoxia.

On the other hand, both the expression of the plasmatic iNOS and levels of NOx significantly decreased at the end of the scheduled workout program. As mentioned above, pulsed hyperoxia induces a NOP response characterized by the activation NF-κB transcription factor which, in turn, triggers the expression of iNOS. Therefore, the observed significant decreases of plasma levels of NOx and iNOS were somehow unexpected. However, elevated levels of iNOS have been reported to be indicative of an inflammatory status [40,41], and our study was conducted on a young population of subjects (mean age: 22.7 ± 2.7 years) in apparent very good health status. In this baseline conditions, it is possible that the hormetic response associated to the activation of NRF2 transcription factor might have prevailed over the induction of iNOS expression, dependent on the NF-κB activity, largely regulated by substrate availability [42]. Moreover, even though the transcription factor NF-κB plays a central role in iNOS regulation, several other factors including TNF-α and IL-6 have been shown to induce iNOS, whereas other factors, including molecules of nutritional interest that were not investigated in the present study, have been shown to inhibit iNOS expression by inhibiting NF-κB activation. In fact, higher levels of NOS require increased levels of tetrahydrobiopterin, and this is consistent with our results and previously reported data [43].

In the present study, we therefore considered the effects of a light exercise program and of modified air composition on selected cytokines, namely IL-6, IL-10, and TNF-α. IL-6 is a pleiotropic cytokine with roles in immunity, tissue regeneration, and metabolism and contributes to host defense during infection and tissue injury, therefore having, in general, a proinflammatory function. On the other hand, the excessive synthesis of IL-6 and the dysregulation of IL-6 receptor signaling are involved in different pathologies [44]. Exercise has been reported to significantly affect Il-6 expression as an “energy sensor” that activates glycogenolysis in the liver and lipolysis in fat tissue in order to provide muscle with the growing energy demands during exercise [45]. At the end of the 5-week workout program, no significant variations of IL-6 were observed in subjects performing in normoxic, light hypoxia, and high hyperoxia conditions. Conversely, a slight significant increase was observed in the group of subjects exercising while breathing a slightly hyperoxic air mixture (30% O_2_). This threshold of the oxygen effect can be explained by the fact that the expression of IL-6 is mainly regulated by the activation of NF-κB [46], which is dependent of oxygen levels. As mentioned above, we have already reported that while mild hyperoxia is associated to the activation of this transcription factor, higher oxygen concentrations induce a response pattern dominated by the activation of NRF2 with no evident upregulation of NF-κB activity [17].

The workout program and different oxygen concentrations also affected IL-10 plasma levels. IL-10 is considered an anti-inflammatory cytokine signaling through a receptor complex consisting of two IL-10 receptor-1 and two IL-10 receptor-2 proteins [47]. Similarly to IL-6 expression, IL-10 expression is regulated by NF-κB and involves ERK1/2 and p38 signaling [48], together with other mechanisms including a negative feed-back loop and an extensive post-transcriptional regulation. In our study, we observed a significant upregulation of IL-10 associated to both hypoxia and mild/high hyperoxia, suggesting that the modification of breathing gas composition during exercise sessions may have an “anti-inflammatory” effect, possibly through distinct mechanisms involving the oxygen-sensing system that finely regulates changes in oxygen availability.

TNF-α plays also an important role in the inflammatory response. It activates the expression of proinflammatory genes in vascular endothelial cells and leukocyte adhesion to vessels, enhancing the infiltration of lymphocyte to the site of infection [49]. TNF-α is often described in the literature as one of the classical NF-κB-dependent proinflammatory cytokines [50]. Therefore, as expected, we observed a significant increase of TNF-α plasma levels in association with the administration of mild hyperoxia during workout sessions. More surprisingly, we observed a significant upregulation of TNF-α also in association with high hyperoxia (100% O_2_), a condition where, as mentioned above, the NF-κB component of the response is absent in favor of a bona fide hormetic response, mediated by NRF-2 activation [35]. This apparent incongruity can be explained thanks to the presence of an NF-κB-independent TNF-α gene transcription, modulated by other transcription factors such as those belonging to the families of STAT and IRF, which have not been considered in our study but already reported to be affected by cellular redox status [51,52].

Our physical activity protocol induced significant variation in biomarkers that have been associated to renal function. However, creatinine and neopterin urinary concentrations are also considered general markers of some metabolic pathways. Creatinine is a by-product of phosphocreatine metabolism. It is produced in a “dose–response” fashion, when phosphocreatine metabolic pathway is engaged by physical activities according to the intensity of exercise [53]. In our study, physical activity performed under hyperoxia was associated with increased creatinine levels at both 30% and 100% pO_2_. In the group of subjects exposed to high hyperoxia, we observed a large spread of individual responses. This high inter-individual variability is possibly due to different subjects’ reactivity to metabolic stressor as the higher oxygen partial pressure. Similarly, exercising while breathing hypoxic air induced a significant increase of creatinine levels. These results match with the increase of Borg’s score, indicating that even a very light physical work load is perceived as “demanding” muscular effort if performed at low oxygen partial pressures, leading to significant creatinine levels [54]. The combination of the RER and Borg scale data suggests that, even though the experimental plan did not lead to a “bona fide” physical training, exercising while breathing a mildly hypoxic air mixture associates with effects at a different level, possibly involving an adaption to fatigue at the level of substrate metabolism [55] and possibly of the Central Nervous System (CNS) [56]. Therefore, it seems possible to speculate that a consistent “metabolic effort” is present when diverse, non-normoxic, oxygen levels are administered in the course of a long-term program based on low-intensity workout.

Neopterin is synthesized by human interferon-gamma-stimulated macrophages and is indicative of a proinflammatory status [57,58]. Increased neopterin concentrations have also been reported in different disorders associated with immune activation and in inflammation [59]. In our study, conducted on a group of apparently healthy young subjects, the exposure to both 30% and 100% hyperoxia during exercise led to significantly increased neopterin levels, paralleling the increase of administered pO_2_. This association suggests the presence of a hormetic response countering the mild oxidative stress triggered by hyperoxia [17] and an activation of the immune system. These data further emphasize that a light-intensity exercise, such as the one considered by our study, does not elicit any significant response in the absence of a hyperoxic stimulus. Therefore, both mild and high hyperoxia act as a “cellular training complement”, making a light exercise comparable to exercise at higher intensities, without an increased challenge for muscularity. Altitude and simulated-hypoxia trainings have been reported to produce different physiological and/or biochemical adaptations in the skeletal muscle [60]. Accordingly, higher-intensity physical exercise in hypoxic environment has been demonstrated to be associated with health benefits of in patients with type 2 diabetes mellitus and to positively change physiological responses in healthy subjects [61]. Conversely, even though few studies on the physiological effects of hyperoxia training have been conducted with conflicting results, a large body of evidence indicates that the oxygen transport capacity, lactate metabolism, power output, and work tolerance (endurance) are improved when breathing hyperoxia during physical exercise [62].

Moreover, considering that, as mentioned above, neopterin is recognized as an expedient biomarker for immune system activation, including monocytes and dendritic cells responsible for the de novo expression of iNOS, our protocols can be assumed to be able to positively impact post viral rehabilitations programs [63,64].

Finally, we also assessed the effect of the long-term exercise program adopting modified oxygen concentrations on urate plasma levels. Urate results from protein breakdown, and therefore, its plasmatic levels are considered reliable markers of the occurrence of an intense muscular effort bout [65]. We found a significantly increase of plasma urate in association with the exposure to hyperoxic air during workout sessions. This increase further suggests that hyperoxia during effort within a 5-week program of light physical exercise elicits metabolic responses similar to that occurring when subjects are challenged with higher impact exercise levels. Noteworthy, similar increased levels of urates have been reported after a single 2 h workout session at 65% of VO_2_ max [65]. Our observation on renal parameters is consistent with the increase of Hb and concomitant hematocrit. In fact, spleen constriction is normally absent during light-to-mild exercise [66] but can be triggered in the presence of hypoxia [67]. Other mechanisms are possibly involved in the “sensing” of hyperoxia and subsequent return to normoxia as a hypoxic stimulus [68].

In conclusion, our study confirmed that mild or high hyperoxia, and at least in part hypoxia have beneficial effects during mild effort, inducing a modulation of the inflammatory status and on the downstream consequences at a systemic level. It should be stressed that the present investigation has been conducted on healthy young subjects with no evidence of altered immunologic or inflammatory parameters. Therefore, our results need to be re-evaluated in a different clinical perspective, possibly designing a clinical intervention study. Finally, the variations observed between hyperoxic and hypoxic challenge and the lack of response of specific markers following the exposure to a mild hyperoxic stimulus indicated that further investigations are needed to establish the best schedule of intermittent hyperoxic stimuli during mild effort to optimize metabolic response.

## 4. Materials and Methods

### 4.1. Experimental Protocol

A total of 26 subjects were included in this study: 6 control subjects performed their exercise program in normoxic conditions, while 7, 6, and 7 subjects followed a similar workout program at 15%, 30%, and 100% O_2_, respectively.

All experimental procedures were conducted in accordance with the Declaration of Helsinki [69] and approved by the Bio-Ethical Committee for Research and Higher Education, Brussels (N° B200-2020-088). After written consent was obtained, 26 healthy, non-smoker subjects with regular but not excessive physical activity—aerobic exercise less than 3 times per week (15 males and 11 females) aged 22.7 ± 2.7 years old (mean ± SD), with heights of 175.3 ± 8.6 m and weights of 70.7 ± 11.4 kg were recruited for this study. They were selected from a large group of physiotherapy students, free of respiratory and cardiac affections as well as pregnancy at inclusion or in the 12 months before the experiment. Once being declared medically fit, they were randomly assigned to one of four groups (one control group and three interventional groups) using an electronic number generator.

The subject assigned to the control group did not undergo any intervention, while the others had to perform a low-intensity exercise every other day to achieve 3 sessions per week for 5 weeks. The exercise consisted of a step-test at a personal given rhythm to achieve a steady power output of 100 W for the entire effort duration. The participants were instructed to step to the beat of a smartphone-based metronome application, using the same lead leg in an up, up, down, and down rhythm during each 4-step cycle. Since the step height (30 cm) was identical for every participant, the pace of the exercise (beat per minute) was based on each subject’s personal weight and tailored according to the following equation modified from [70] ($\frac{1}{\frac{(Weight\;\left(kg\right)*9.81*0.3}{100}}*240)\;$ to achieve the same effort during the same time laps. This exercise protocol was provided to fulfill some rehabilitation programs after detraining periods for different reasons.

Each test had a duration of 20 min, during which each participant blindly breathed a different mixture, either 15%, 30%, or 100% of inspired oxygen (pO_2_ of 150 hPA (group 1), 300 hPA (group 2), and 1000 hPA (group 3), respectively). In groups 1 and 2, oxygen was administered by means of an orofacial non-rebreather mask with reservoir tightly fit on the subjects’ faces. The breathing gas flow from a pressurized gas tank was set at 10 L·min^−1^. Group 3 received 100% of oxygen from an oxygen concentrator (NewLife intensity, CAIRE Inc, Ball Ground, GA, USA) with a similar non-rebreathing mask set-up.

Before and after the exercise session, each participant performed a cycle ergometer stress test to determine ventilatory parameters during effort along with blood and urine samples. After each exercise session, the subjects assessed Borg-perceived exertion ratings for both respiratory and leg discomfort.

### 4.2. Cycle Ergometer Stress Test

The stress test was performed on an electro-mechanically braked ergometer that required the subjects to maintain a cycling rate of 60 to 65 RPM to keep the work rate constant. The stress test consisted of a graded exercise including a 3-min 50 W warm-up followed by short incremental steps (25 W·min^−1^) up to a maximum of 300 W. The stress test was conducted until volitional exhaustion, followed by a 3-min active recuperation at the same cycling rate.

During exercise, the subjects were monitored by 3-lead electrocardiogram (ECG), pulse oximetry, and blood pressure was taken by means of a manual sphygmomanometer. They breathed through a mouthpiece with a nose clip in place. Oxygen uptake (VO_2_), carbon dioxide output (VCO_2_), and minute ventilation (VE) were measured breath by breath (Exp’air R57.2, Medisoft, Sorinnes, Belgium). To guarantee the system accuracy, the airflow and gas concentrations were calibrated prior to each test. Ventilatory thresholds 1 (VT1) and 2 (VT2) were determined using the V-slope method (VCO_2_/VO_2_) [67] as the primary criterion and the VE/VCO_2_ method [71] was used as the secondary criterion. The individual anaerobic threshold (LAT) was determined according to VT1 and VT2 [72]. The RER was defined as the VCO_2_/VO_2_ ratio.

### 4.3. Venous Blood Samples Analysis

Approximately 10 mL of venous human blood were drawn from an antecubital vein, with subjects sitting or lying on a bed. Samples were collected in lithium heparin and EDTA tubes (Vacutainer, BD Diagnostic, Becton Dickinson, Italia S.p.a.). Plasma and red blood cells (RBCs) were separated by centrifugation (Eppendorf Centrifuge 5702R, Darmstadt Germany) at 1000× *g* at 4 °C for 10 min. The samples were then stored in multiple aliquots at −80 °C until assayed; analysis was performed within one month from collection.

#### 4.3.1. Redox State

Briefly, an electron paramagnetic resonance instrument (E-Scan—Bruker BioSpin, GmbH, Rheinstetten, Germany) X-band was adopted for determinations. A controller ‘‘Bio III’’ unit (Noxygen Science Transfer & Diagnostics, Elzach, Germany), interfaced to the spectrometer, was used to stabilize the samples temperature at 37 °C.

The ROS production rate and the TAC were determined as already performed by some of the authors on plasma [29,73,74,75,76,77].

Briefly, for ROS detection, 50 μL of each plasma sample were immediately treated with 1-hydroxy-3-methoxy-carbonyl-2,2,5,5-Tetramethylpyrrolidine (CMH) probe (*v*:*v*, 1:1). Fifty microliters of the obtained solution were put in a glass EPR capillary tube and in turn placed inside the spectrometer cavity for data acquisition. A stable radical (3-carboxy2,2,5,5-tetramethyl-1-pyrrolidinyloxy (CP) was used as an external reference to convert ROS determinations in absolute quantitative values (μmol·min^−1^). While for TAC assessment, a 2,2-diphenyl-1-picrylhydrazyl (DPPH•) spin trap was used. Five microliters of plasma were added to 45 μL of a buffer solution; then, the reaction was initiated by the addition of 50 μL of DPPH• (mM DPPH• in absolute ethanol) as a source of free radicals, as previously indicated [62,78]

All steps were performed in the dark to avoid photochemical effects on DPPH•. A linear calibration curve was computed from pure Trolox-containing reactions; TAC was expressed in terms of trolox-equivalent antioxidant capacity (mM). All spectra were analyzed in duplicate by using Win EPR software (2.11 version, Bruker BioSpin, GmbH Rheinstetten, Germany) standardly supplied by Bruker.

#### 4.3.2. Inflammatory Status

IL-6, IL-10, and TNF-α plasmatic levels were measured by human interleukins ELISA kits (Cayman Chemical, Ann Arbor, MI, USA) according to the instructions. All samples and standards were read by a microplate reader spectrophotometer (Infinite M200, Tecan Group Ltd., Männedorf, Switzerland). The determinations were assessed in duplicate, and the inter-assay coefficient of variation was in the range indicated by the manufacturer.

#### 4.3.3. Nitric Oxide Measurements

Nitrite and Nitrate (NOx) concentrations were determined on urine with a colorimetric method based on the Griess reaction, using a commercial kit (Cayman Chemical, Ann Arbor, MI, USA) as previously described [76,79].

iNOS expression in plasma was assessed by a human NO_2_/iNOS ELISA kit (catalogue no.: EH0556; FineTest, Wuhan, China). This assay was based on the sandwich enzyme-linked immune-sorbent assay technology.

All the samples were read in duplicate, with each at a correct wavelength by a spectrophotometer microplate reader (Infinite M200, Tecan Group Ltd., Männedorf, Switzerland), according to the manufacturer’s instructions.

### 4.4. Urine Sample Analysis

Urine was collected by voluntary voiding in a sterile container and stored in multiple aliquots at −20 °C until assayed and thawed only before analysis.

Urinary creatinine and neopterin concentrations were measured by the high-pressure liquid chromatography (HPLC) method as previously described [80]. In addition, uric acid levels were determined by Varian instrument (pump 240, autosampler ProStar 410) coupled to a UV-VIS detector (Shimadzu SPD 10-AV, λ = 240 nm) after centrifugation at 13,000 rpm at 4 °C for 5 min. Analytic separations were performed at 50 °C on a 5 µm Discovery C18 analytical column (250 × 4.6 mm I.D., Supelco, Sigma-Aldrich) at a flow rate of 0.9 mL·min^−1^. The calibration curves were linear over the range of 0.125–1 μmol·L^−1^, 0.625–20 mmol·L^−1^, and 1.25–10 mmol·L^−1^ for neopterin, uric acid, and creatinine levels, respectively. The inter-assay and intra-assay coefficients of variation were <5%.

### 4.5. Statistical Analysis

The normality of data was performed by means of Shapiro–Wilk tests. When a Gaussian distribution was assumed, they were analyzed with a one-way ANOVA for repeated measures with Dunnett’s post hoc test; when comparisons were limited to two samples, paired or non-paired *t*-tests were applied. If the Gaussian distribution was not assumed, the analysis was performed by means of a non-parametric multiple comparisons—Dunn’s test. Taking the baseline measures as a reference (100%), percentage variations were calculated for each condition, allowing an appreciation of the magnitude of change rather than absolute values. All statistical tests were performed using a standard computer statistical package, GraphPad Prism version 5.00 for Windows (GraphPad Software, San Diego, CA, USA). A threshold of *p* < 0.05 was considered statistically significant. All data are presented as mean ± standard deviation (SE).

## Figures and Tables

**Figure 1 ijms-22-09600-f001:**
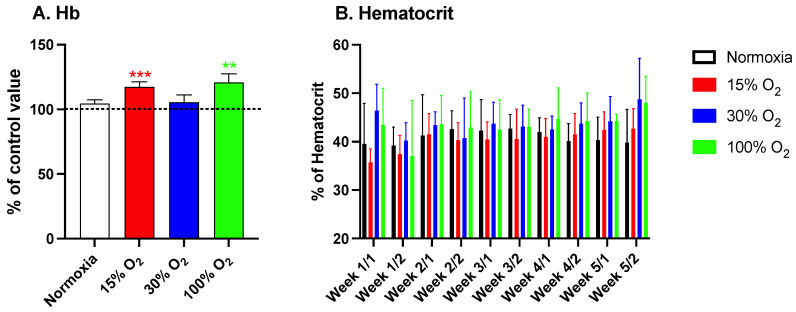
Hemoglobin variation after a 5-week exercise program consisting of 20 min low-intensity workout sessions every other day (panel (**A**)) and hematocrit variation throughout the same program (panel (**B**)). Histograms display the percentage variation (mean ± standard deviation (SE)) in comparison to the baseline (time 0). ** *p* < 0.01, *** *p* < 0.001; for one sample *t*-test vs. the baseline. Samples were drawn twice a week.

**Figure 2 ijms-22-09600-f002:**
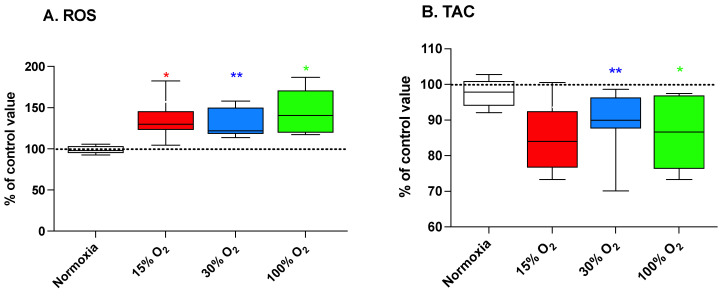
Reactive oxygen species (ROS) production (panel (**A**)) and total antioxidant capacity (TAC; panel (**B**)) in human plasma at the end of a 5-week workout program consisting of 20 min low-intensity exercise sessions every other day. Box and Whisker plots indicate the median, 1st quartile, 3d quartile, interquartile range, minimum, and maximum in comparison to the baseline (time 0), which was set at 100%. * *p* < 0.05, ** *p* < 0.01 vs. the baseline before oxygen exposure (time 0), according to the Wilcoxon test.

**Figure 3 ijms-22-09600-f003:**
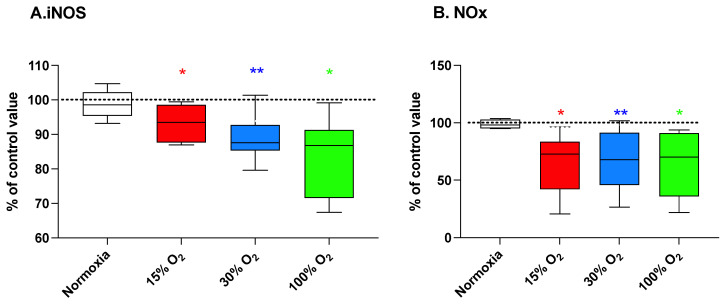
Inducible nitric oxide synthase (iNOS) protein level (panel (**A**)) and NOx production (panel (**B**)) in human plasma at the end of a 5-week workout program consisting of 20 min low-intensity exercise session every other day. Box and Whisker plots indicate the median, 1st quartile, 3d quartile, interquartile range, minimum, and maximum in comparison to the baseline (time 0), which was set at 100%. * *p* < 0.05, ** *p* < 0.01 vs. the baseline before oxygen exposure (time 0), for the Wilcoxon test.

**Figure 4 ijms-22-09600-f004:**
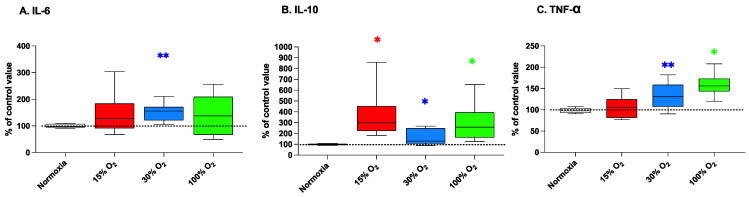
Interleukin (IL)-6 (panel (**A**)), IL-10 (panel (**B**)), and tumor necrosis factor-alpha (TNF-α) (panel (**C**)) in human plasma after a 5-week workout program consisting of 20 min low-intensity exercise sessions every other day. Box and Whisker plots indicate the median, 1st quartile, 3d quartile, interquartile range, minimum, and maximum in comparison to the baseline (time 0), which was set at 100%. * *p* < 0.05, ** *p* < 0.01 vs. the baseline before altered oxygen exposure (time 0), according to the Wilcoxon test.

**Figure 5 ijms-22-09600-f005:**
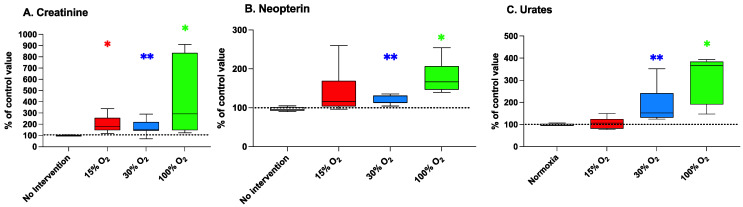
Creatinine (panel (**A**)), neopterin (panel (**B**)), and urates (panel (**C**)) in human urine at the end of a 5-week workout program consisting of 20 min low-intensity exercise sessions every other day. Box and Whisker plots indicate the median, 1st quartile, 3d quartile, interquartile range, minimum, and maximum in comparison to the baseline (time 0), which was set at 100%. * *p* < 0.05, ** *p* < 0.01 vs. the baseline before oxygen exposure (time 0), for the Wilcoxon test.

**Figure 6 ijms-22-09600-f006:**
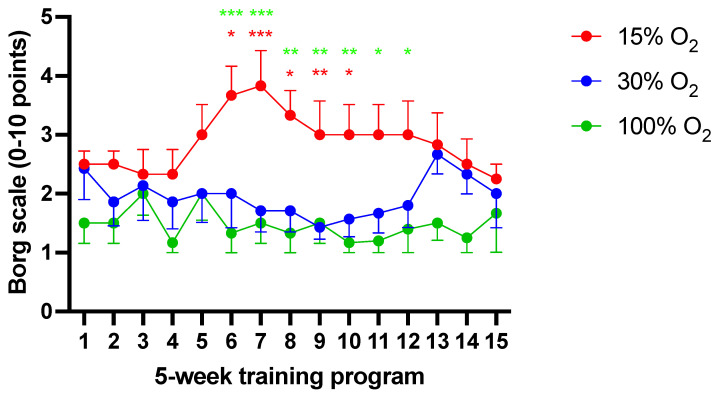
Longitudinal evaluation of Borg’s perceived excretion rating (mean ± SEM). * *p* < 0.05, ** *p* < 0.01, *** *p* < 0.001; for the Kruskal–Wallis test followed by Dunn’s post hoc test. (red color: 15% vs. 30%; green color: 15% vs. 100%).

## Data Availability

Data are available at request from the authors.

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
