# Peer review of "Hypoxic and Hyperoxic Breathing as a Complement to Low-Intensity Physical Exercise Programs: A Proof-of-Principle Study"

_ijms, 2021, doi:10.3390/ijms22179600_

Round 1

Reviewer 1 Report

The topic of this manuscript falls within the scope of International Journal of Molecular Sciences. The topic of the manuscript is very interesting, relevant, and original. hypoxia has beneficial effects during mild effort inducing a modulation of the inflammatory status. The  data has been provided with vigorous statistical analysis. The Authors have presented sufficient data. The appropriate tables and figures have been provided. The article is easy to read and logically structured. The conclusions are consistent with presented evidence and arguments. References are up to date and complete. In my opinion this paper  may be published in present form.

Author Response

topic of the manuscript is very interesting, relevant, and original. hypoxia has beneficial effects during mild effort inducing a modulation of the inflammatory status. The  data has been provided with vigorous statistical analysis. The Authors have presented sufficient data. The appropriate tables and figures have been provided. The article is easy to read and logically structured. The conclusions are consistent with presented evidence and arguments. References are up to date and complete. In my opinion this paper may be published in present form.

Authors really appreciate the positive comments by this referee. Receiving positive feedback encourage us to foster our willingness in pursuing our research.

Reviewer 2 Report

This is an interesting proof-of-concept paper on the adaptation responses developed by healthy young individuals to a 5-week low intensity exercise program performed under hyperoxia or mild / high hyperoxia. The present work derived from previous studies on the changes triggered by intermittent exposure to various conditions of partial oxygen pressure. Such alternative therapeutic approaches are highly needed as single therapy or co-therapy for modulating immune responses in chronic diseases.

Going beyond making proof-of-concept, the main issue is that the study was focused on healthy young individuals, and it is not clear how the observed changes are impacting in the short and long run on immune and redox homeostasis. Moreover, it is not at all obvious that the proposed therapeutic approach would work in some pathologic conditions (they should be mentioned in correlation with the changes registered in the study). For instance, the approach might work in young healthy individuals with moderate immunodeficiency (not the case in the study) as preventive therapy in COVID-19, but an increase of the levels of pro-inflammatory cytokines might not be beneficial during the disease when a cytokine storm might be induced by the virus. Additionally, would a moderate increase of the levels of pro-inflammatory cytokines be beneficial for chronic non-infectious diseases that are most of them underlined by low-grade inflammation and oxidative stress? Difficult to say based on the existing results.  

These limitations of the study were mentioned in the paper, but I consider that, from the beginning, the author should point out that this is a proof-of-concept study.

Another issue, in the light of making proof-of-concept, is that the mechanism of the observed changes should be investigated at molecular level, at least regarding the transcriptional activity of NF-kB and HIF-1α in blood leukocytes. It would be also useful for the demonstration to address some target genes of the transcription factor NRF2 which is known to have a crosstalk with NF-kB and HIF-1α, and has a crucial role in maintaining redox homeostasis.

The paper is very well written, but I suggest some improvement of clarity as follows. While results are structured on individual parameters in all the investigated conditions, the discussion is better to be focused on each investigated condition, otherwise it is difficult to follow.

Other comments:

  • In Figure 1, please comment on panel B.
  • In the Method section, please give briefly some information on the oxidative and antioxidant activity that is assessed by ESR using particular spin traps.
  • You mentioned several times that the low intensity exercise in hyperoxic conditions gives almost the same results as heavier exercise in normoxic conditions, but you did not have a group performic only heavy exercise in the present study. At least you should give a reference where these data can be found.
  • You mentioned several times that “Similarly, when the work-out program was performed at normal pO2 no significant changes of … were observed”. Data should be provided, eventually in a supplementary file.
  • Please revise lines 206-208 regarding creatinine levels. In Figure 5 you show statistical differences at 100% O2, but in the text you mention only 15% and 30%. Moreover, should clearly mention in the text to which condition the data are referring to.

You can find some other comments in the attached manuscript.

Author Response

These limitations of the study were mentioned in the paper, but I consider that, from the beginning, the author should point out that this is a proof-of-concept study.

We agree with all the points raised by the referee. Having recruited young healthy subjects rather than “real” patients can be considered a limitation of our study. Nonetheless, testing our hypothesis straightforward on diseased individuals, irrespective to the seriousness of their impairment, would have been somehow “unethical”, as both hyperoxic, either mild or frankly, and hypoxic treatment are a challenging stress. Any ethical committee would have negatively considered (with reasons) our protocols without the support of some preliminary indication of efficacy. This was indeed the basic idea underlying our study: to provide a background for further studies, directly addressing the effect of air composition in specific population of individuals.
The text has been modified according to this comment. The “proof-of-concept” nature and the aim of our study are now better highlighted.

Another issue, in the light of making proof-of-concept, is that the mechanism of the observed changes should be investigated at molecular level, at least regarding the transcriptional activity of NF-kB and HIF-1α in blood leukocytes. It would be also useful for the demonstration to address some target genes of the transcription factor NRF2 which is known to have a crosstalk with NF-kB and HIF-1α, and has a crucial role in maintaining redox homeostasis.

Indeed, we totally concur with the reviewer’s point of view. However, please note that we have already investigated about the effect of pulsed hypoxia and hyperoxia, applying an almost identical experimental protocol, on the activation of NF-κB, HIF and NRF2 transcription factors in blood leukocytes! Our results have been recently published by the same journal to which we are submitting the present study (see https://www.mdpi.com/1422-0067/22/1/458/pdf). Noteworthy, the previous study was performed on a group of subjects with characteristics overlapping those of the participants recruited into the present one. Therefore, re-considering these specific molecular events has been considered not novel and possibly “useless”.

The paper is very well written, but I suggest some improvement of clarity as follows. While results are structured on individual parameters in all the investigated conditions, the discussion is better to be focused on each investigated condition, otherwise it is difficult to follow.

In the light of the style adopted in the discussion section of the previous paper (see the rebuttal above), we wanted to apply a similar structure to the present manuscript. We made this choice in the hope that this could help the readers to line-up the two sets of results, according to the observed effects rather than to treatments. Therefore, even though we have appreciated and considered the suggestion made by the referee, we elected to leave the structure of the discussion unchanged.

Other comments:

  • In Figure 1, please comment on panel B.

A comment to the panel B of Figure 1 has been added.

  • In the Method section, please give briefly some information on the oxidative and antioxidant activity that is assessed by ESR using particular spin traps.

We originally wanted to keep the length of the methods section limited, considering that more information can be provided from the original papers cited were appropriate. However, we harvested the comment by the referee and some lines were added to provide more details about the analytical methods utilized.

  • You mentioned several times that the low intensity exercise in hyperoxic conditions gives almost the same results as heavier exercise in normoxic conditions, but you did not have a group performing only heavy exercise in the present study. At least you should give a reference where these data can be found.

We did not plan a group undergoing a heavy (training) exercise program under either hypoxic or hyperoxic conditions as these information have already been reported by others. However, we agree with the point raised by the reviewer and specific references have been added where appropriate in the discussion section.

  • You mentioned several times that “Similarly, when the work-out program was performed at normal pO2 no significant changes of … were observed”. Data should be provided, eventually in a supplementary file.

We are not quite much sure we understand which supplemental data are suggested to be added. Actually, all graphs report the effects of hypoxic and hyperoxic breathing while exercising in comparison to the “normobaric” group which is also plotted as “normoxia”. Data utilized in these plots refer to the difference between initial values (observed before the beginning of the program) and final values (assessed at the end of the 5 weeks work-out program). Indeed, all graphs clearly show that exercising under normoxic air (sham conditions as mentioned in the methods) did not result in any significant variation of the parameters considered.

  • Please revise lines 206-208 regarding creatinine levels. In Figure 5 you show statistical differences at 100% O2, but in the text you mention only 15% and 30%. Moreover, should clearly mention in the text to which condition the data are referring to.

The text has been modified accordingly.

  • You can find some other comments in the attached manuscript.

All comments included in the manuscript have been considered. Moreover, following some of the reviewer's comments we also decided to change the title for a more appropriate one.

Round 2

Reviewer 2 Report

The authors responded to the issues raised at the first manuscript reviewing.

The article has consequently gained in clarity.